# A functional vulnerability framework for biodiversity conservation

Arnaud Auber [1] ✉, Conor Waldock[2,3], Anthony Maire [4], Eric Goberville [5], Camille Albouy[6,7], Adam C. Algar [8], Matthew McLean [9], Anik Brind'Amour[10], Alison L. Green[11], Mark Tupper[12,13], Laurent Vigliola [14], Kristin Kaschner [15], Kathleen Kesner-Reyes [16], Maria Beger[17,18], Jerry Tjiputra [19], Aurèle Toussaint [20], Cyrille Violle[21], Nicolas Mouquet[22,23], Wilfried Thuiller[24] & David Mouillot [23,25]

Setting appropriate conservation strategies in a multi-threat world is a challenging goal, especially because of natural complexity and budget limitations that prevent effective management of all ecosystems. Safeguarding the most threatened ecosystems requires accurate and integrative quantification of their vulnerability and their functioning, particularly the potential loss of species trait diversity which imperils their functioning. However, the magnitude of threats and associated biological responses both have high uncertainties. Additionally, a major difficulty is the recurrent lack of reference conditions for a fair and operational measurement of vulnerability. Here, we present a functional vulnerability framework that incorporates uncertainty and reference conditions into a generalizable tool. Through in silico simulations of disturbances, our framework allows us to quantify the vulnerability of communities to a wide range of threats. We demonstrate the relevance and operationality of our framework, and its global, scalable and quantitative comparability, through three case studies on marine fishes and mammals. We show that functional vulnerability has marked geographic and temporal patterns. We underline contrasting contributions of species richness and functional redundancy to the level of vulnerability among case studies, indicating that our integrative assessment can also identify the drivers of vulnerability in a world where uncertainty is omnipresent.

Further climate change is inevitable[1], but other pressing environmental and ecological threats are also widespread and intensify on Earth[2], including land use changes, pollution, species invasions, diseases and resource overexploitation. These human-based impacts pose imminent hazards to people and species that compose and sustain ecosystems[3–5]. Quantifying the vulnerability of biodiversity—i.e., the degree to which biodiversity and associated functions are likely to change when exposed to multiple threats[6]—is thus crucial[2,7] to rationalize ecosystem management and conservation actions[8]. Such a

quantification is especially needed in the current context of budget limitations while new global $CO_2$ emission and conservation targets are being set[9]. Despite major advances in our understanding of ecosystem vulnerability, shortcomings persist[10–13]. For example, the majority of studies only estimate vulnerability based on correlative species-environment relationships or expert judgements[14]. Further, vulnerability assessments generally focus on specific disturbances such as fishing[15], flood events[16], fire[17], trawling[18], wind farms[19] or climate change[20]. However, the complexity of biological responses and the

multi-threat nature of the Anthropocene demands conservation scientists to consider a wide range of disturbances in a more generalizable, standardized, and operational framework to assess ecosystem vulnerability[21–23].

One of the main difficulties in ecosystem health assessment is the need for benchmarks, most often called 'reference conditions', to evaluate the status of ecosystems based on indices[24–26]. A reference condition is broadly defined as a baseline measure of an ecosystem variable (biological, chemical, or physical attributes) representative of minimal human influence or stress[24,25,27]. Despite the intuitive relevance of this concept for ecological assessment, the majority of ecological measures—including community-scale indices—are rarely compared to reference conditions, preventing end-users from fairly and operationally gauging impacts or restoration actions. Considering reference conditions to assess the vulnerability of any observed community therefore appears as a necessary step: it allows situating community's response to disturbances relative to the widest range of possibilities, i.e., from the 'least' to the 'most' vulnerable community. Such conditions—or quasi-pristine areas—are now virtually absent for most ecosystems on Earth[28,29] however, and past data about pre-human conditions are often biased and limited[27]. Moreover, the natural stochasticity in the Earth's climate (e.g., volcanism, extreme meteorological events), the lack of data relative to the combined effects of various disturbances on biodiversity, and the extreme complexity of biological responses themselves[30], still prevent accurate predictions of species community shifts under global changes[31,32]. Such unpredictability hampers our ability to cope with future environmental and socioeconomic changes, weakening management and conservation efforts. Assessing vulnerability therefore requires estimating community sensitivity and exposure by considering a wide range of disturbances rather than focusing only on a specific threat[21–23].

Examining the diversity of organismal traits instead of their taxonomic classification can provide a more mechanistic understanding of community dynamics[33–35], including their resilience to various environmental conditions and human impacts[7,36]. Here, we assume that the 'functional vulnerability' of a community to a wide range of disturbances relies on two components: (i) functional redundancy[36,37], i.e. the extent to which different species share similar traits and are thus likely to sustain similar functions, and (ii) how species respond to different disturbances[15,36]. However, community functional redundancy is still often ignored in functional vulnerability assessments while responses to multiple threats are poorly known. Ecological theory predicts that communities composed of species with low redundancy across multiple traits may be more vulnerable to multiple disturbances since having only few functionally similar species, if not one, responding in diverse ways to these disturbances. In contrast, high functional redundancy is expected to provide many species per threat-response, i.e., a more even distribution of species redundancy across traits, which may buffer functional extinction risk[38]. Moreover, functionally distinct species are known to be highly vulnerable to many pressures such as overexploitation[39], habitat loss[40] and climate change[41]. Considering species traits, including their distinctiveness[42], and their abundance, is therefore a necessary step in assessing the functional vulnerability of communities to multiple threats. Yet, a framework that considers the irreducible uncertainty and complexity of nature in terms of current and future environmental disturbances as well as species responses is still missing.

Here, we develop an integrative trait-based framework to quantify the functional vulnerability of biological communities to a wide set of potential disturbances while taking into account extreme cases of community responses as reference conditions. Based on species position, species abundances redistribution in the functional trait space and in silico simulations of disturbances, our framework considers functional rarity, redundancy and species abundances to quantify vulnerability in a multi-threats context. This study shows that our framework constitutes a promising tool to guide protection efforts even if trait-environment relationships and disturbance regimes are unknown, unpredictable or poorly documented. To test the applicability of our framework in both time and space, we consider three contrasting marine case studies: the past temporal dynamics of species abundances in North Sea fish communities, the current occurrences of marine mammals at global scale, and the projections of reef fish communities at global scale according to future climate.

## Results

### Functional vulnerability framework

To estimate the functional vulnerability of a given community, our framework estimates its capacity to maintain its set of traits using in silico simulations of random disturbances. More precisely, we compared the response of a given observed community with the response of virtual communities built from the observed community itself. These virtual communities are based on changes in (i) the distribution of functional redundancy across the trait space, (ii) abundance distribution across species, and (iii) relationship between species abundance and functional distinctiveness (Fig. 1a), all of which are key aspects of ecosystem functioning and resilience to disturbance. This results in a set of 15 virtual communities (Fig. 1a). Even if the objective of the framework is not to identify which community characteristic determines functional vulnerability, we expect the functional vulnerability of communities to decrease with (i) species redundancy across the trait space, (ii) more balanced abundances across species and (iii) positive relationship between species abundance and functional distinctiveness.

For any observed community, we built a regular grid cell on a two-dimensional trait space where species are located according to their traits[43], the limits of the grid being defined by the upper, lower, leftmost and rightmost species in the trait space. We defined each cell as a 'functional entity' since the position in the trait space can be considered a proxy of species roles in ecosystems. This framework could be extended to multidimensional spaces, where the functional entities would then be defined as hypercubes. We then simulated a series of disturbances on the observed community and on each of its associated virtual communities. For each disturbance, we applied a decrease in the abundance of randomly selected species and recalculated new abundances. The number of functional entities, or grid cells, with at least one species still present was then quantified. These simulated disturbances were applied successively until total species and functional entity extinction (see rarefaction curves, Fig. 1b). The functional vulnerability index was then computed by quantifying the position of the rarefaction curve corresponding to the observed community in comparison to the most vulnerable virtual community and the least vulnerable virtual community (Fig. 1b, c).

To test the applicability of our framework in both time—e.g., recovery after abrupt and intense disturbance—and space—e.g., spatial prioritization for ecosystem management—we considered three contrasting marine case studies.

### Temporal vulnerability of North Sea fishes

We first applied our functional vulnerability framework on the past temporal dynamics of North Sea fishes (Fig. 2a). The North Sea has been subject to intense overfishing throughout the 20th century but has been managed since the end of the 1970s by the Common Fisheries Policy, with a progressive decrease in catch quotas and improvement in gears' selectivity. We used abundance data from the International Bottom Trawl Survey (IBTS), an annual monitoring campaign designed to monitor fish and invertebrate communities in

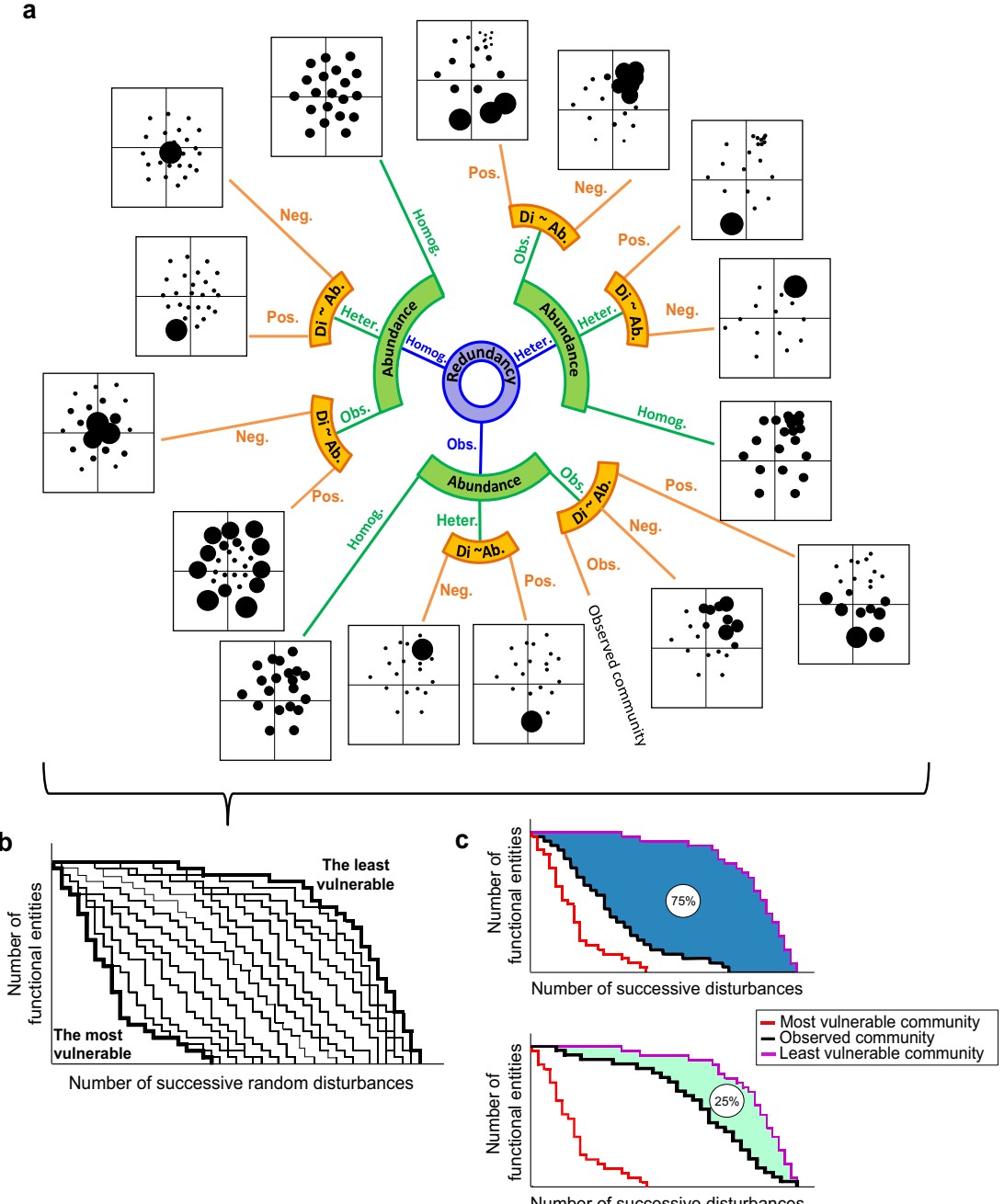

**Fig. 1 | Conceptual representation of the functional vulnerability framework.**
**a** The various trait spaces correspond to the virtual communities created from the observed target community. The size of each dot (i.e., species) is proportional to abundance (Ab.). The blue circle separates virtual communities into three main categories according to the distribution of functional redundancy: heterogeneous (Heter.), homogeneous (Homog.), observed (Obs.). The green and orange circle segments subdivide the virtual communities according to the distribution of species' abundance (heterogeneous, homogeneous, observed) and the positive (Pos.) or negative (Neg.) sign of the relationship between functional distinctiveness

(Di.) and abundance, respectively. **b** Conceptual figure showing the rarefaction curves of the observed target community and its associated virtual communities. The number of randomly selected species at each disturbance does not influence the final outputs, but impacts the number of successive disturbances needed to reach only one species left in the community. **c** Examples of rarefaction curves of two contrasting communities according to their functional vulnerability (25% and 75%). The red and violet curves correspond to the rarefaction curve of the most and least vulnerable communities, respectively. The black curve corresponds to the rarefaction curve of the observed community.

the North Sea annually since the 1980s (Fig. 2a) and to assess commercial species stocks. In combination with species abundance, and for each fish taxa, we selected eight traits linked to ecosystem functioning: age at maturity, asymptotic length, fecundity, offspring size, habitat (water column position), trophic level, feeding mode, and spawning type.

Our framework revealed a high functional vulnerability of fish communities in the North Sea. The functional vulnerability of

observed communities was very close to that of the most vulnerable virtual community, with a value of 90% on average. We found a significant decrease in functional vulnerability throughout the last four decades of about 1.1% per decade (Pearson's correlation test, $r = -0.79$, $P < 0.001$, $n = 36$), dropping from 92 to 86% (Fig. 2b). During the same period, species richness increased from 74 species in the 1980s to 117 species in the 2010s. The analysis also revealed that whatever the number of iterations performed, the vulnerability

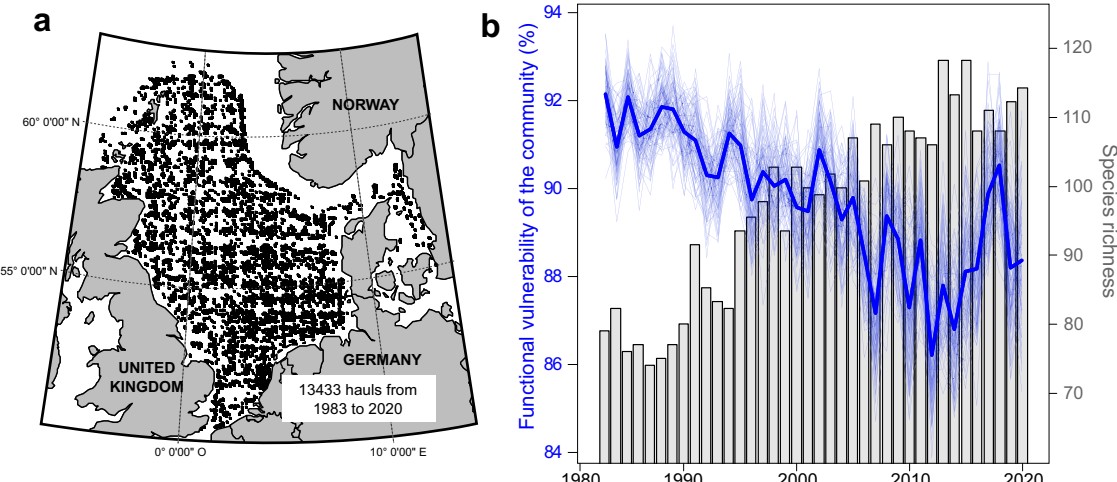

**Fig. 2 | Sampling effort of the IBTS survey and temporal dynamics of vulnerability in the North Sea fish communities. a** Spatial and temporal coverage of the North Sea International Bottom Trawl Survey (NS-IBTS) from 1983 to 2020 (each point represents a 30 min haul). The background map was obtained through the *maps* r package (v.3.3.0). **b** Temporal dynamics of fish functional vulnerability and species richness in the North Sea from 1983 to 2020. The dark blue line denotes the mean functional vulnerability and the light blue lines represent each of the 99 iterations. Source data are provided as a Source Data file.

index remained almost unchanged (relative SD = 0.75%; see variation between light blue lines in Fig. 2b), indicating a high robustness in functional vulnerability assessment. The intra-year variability of functional vulnerability (i.e., the standard deviation of the 99 vulnerability values at each year; see Fig. 2b) was on average two times inferior to the inter-year variability, therefore indicating a good capacity of the framework to detect small temporal changes.

### Global patterns of marine mammals vulnerability

To test our framework on occurrence data, we used global range maps of 122 marine mammal species (http://www.iucnredlist.org). We transformed these range maps into a presence/absence matrix by overlaying species' range maps and summed the number of species that occurred in each 1° grid cell[20]. Finally, we collected fourteen traits usually included in climate change vulnerability assessments, i.e. diet, foraging water depth, foraging location, fasting strategy, terrestriality, female sexual maturity, weaning, gestation length, inter-litter interval, breeding location, social group size, social behavior, adult body mass, sexual dimorphism (see ref. 20 for further details).

We detected a latitudinal gradient in the functional vulnerability of marine mammal communities, with the most vulnerable communities living at northern temperate latitudes (Fig. 3a, c). Opposite to the North Sea case study, we observed a weak but noticeable positive relationship between species richness and functional vulnerability for marine mammals (slope = 1.68; Pearson's correlation test, $r = 0.32$, $P < 0.001$, $n = 19681$; Fig. 3b, d). This result suggests that species-rich communities may have lower trait redundancy across functional entities than species-poor communities. However, some species-poor regions display high functional vulnerability such as the eastern Mediterranean Sea, the Red Sea and the northeast Pacific. Similarly, certain areas—like Indonesian Islands or the South African coast—with high species richness have very low functional vulnerability, therefore supporting the expectation that species-rich communities may have, on average, lower functional vulnerability. Finally, we observed a substantial variation in functional vulnerability values in species-rich communities, starting around 25 species (Fig. 3d) suggesting a species richness threshold above which functional redundancy counterbalances the buffering effect of species richness on vulnerability.

### Future changes in reef fish vulnerability

We finally estimated how climate change could affect the global patterns of functional vulnerability in shallow-water reef fishes over the coming 60 years (2041–2070 compared to the period 1981–2015). To do so, we used species distribution models[44] (see supplementary information) to forecast habitat suitability for 2320 species after integrating >12 million records of species occurrences extracted from open access databases (GBIF[45], OBIS[46]) and local SCUBA transects (Reef Life Survey[47,48], SERF[49], GASPAR[50]). We used the SSP1-2.6 climate change scenario (Shared Socioeconomic Pathways combined with the Representative Concentration Pathway 2.6[23]) that is inherent to the Paris Agreement that aims to sustainably limit global warming to less than 2 °C[51]. We chose twelve fish traits: age and length at maturity, maximum age, the asymptotic mass and length, the Von Bertalanffy growth coefficient (from FISHLIFE[52]), mobility, period of activity, schooling, vertical position, diet, and reef association (from the GASPAR traits database[50]).

Under the 2 °C warming scenario (i.e. SSP1-2.6), the forecasted functional vulnerability of fish communities on shallow-water coral and rocky reefs for the period 2041–2070 is expected to be lower in the tropics (Fig. 4a) where species richness is expected to remain high (Supplementary Fig. 11). As for the North Sea (case study 1), functional vulnerability is expected to be negatively related to species richness (Fig. 2b), yet with a threshold in species richness -of about 800 species- above which functional vulnerability rebounds towards higher values (Fig. 4b). As for marine mammals (case study 2), the distribution of functional redundancy in coastal fish communities, as well as their abundance distribution, counterbalanced the buffering effect of species richness above a certain richness threshold. Future functional vulnerability hotspots are identified around Southern Latin America, South Africa, Northern Hudson Bay, Nova Scotia and the Bering Sea (Fig. 4a). Conversely, the least vulnerable communities are located in Easter Islands, Northeast of French Polynesia and Saint Helena in the Southern Atlantic (Fig. 4a).

The global vulnerability of reef fish communities is expected to increase by 3 ± 10% on average until the end of the 21st century. Around 65% of the considered area is expected to experience an increase in functional vulnerability, with a mean increase of about 12%. Likewise, regions with decreasing functional vulnerability would experience a 11% decrease on average. Globally, polar areas, and

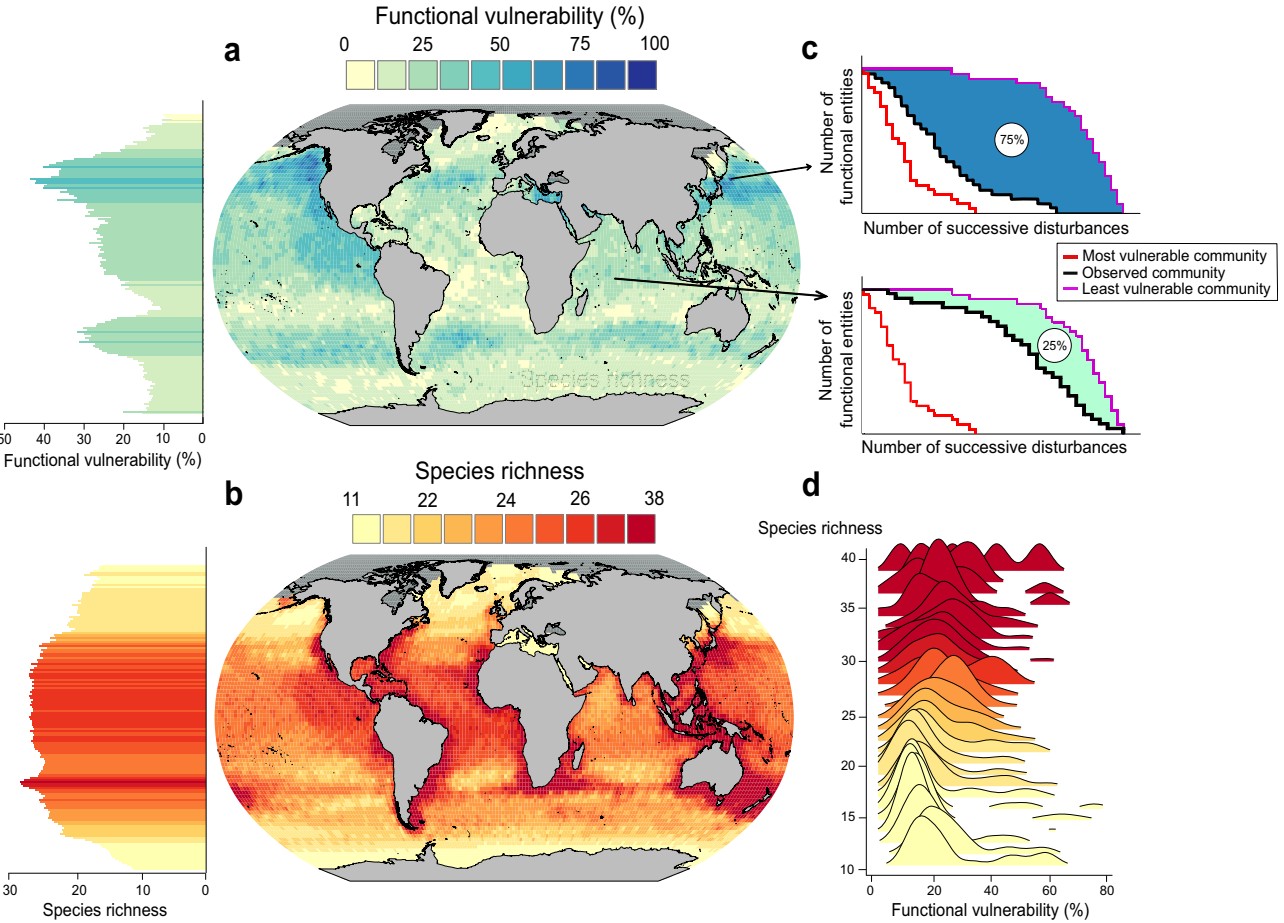

**Fig. 3 | Functional vulnerability and species richness in marine mammal communities. a** Functional vulnerability of marine mammal communities at global scale. **b** Species richness of marine mammal communities. **c** Rarefaction curves of two contrasting communities according to their functional vulnerability (25 and 75%). The red and violet curves correspond to the rarefaction curve of the most and least vulnerable communities, respectively. The black curve corresponds to the rarefaction curve of the observed community. **d** Distribution of functional vulnerability values along the species richness gradient. Sites for which species richness was inferior to 10 were not considered to prevent any over or underestimation of functional vulnerability. Source data are provided as a Source Data file. Background map shapefiles are available on the NOAA website: https://www.ngdc.noaa.gov/mgg/shorelines/data/gshhg/latest/.

especially in the Northern hemisphere, would benefit from a decrease in functional vulnerability whereas tropical, subtropical and some temperate latitudes would suffer from an increase in functional vulnerability over the coming decades (Fig. 4c). We also reveal the existence of certain peculiar locations like the Mediterranean Sea, Latin America and Africa (Fig. 4c) where functional vulnerability is expected to decrease.

## Discussion

Decision-makers are forced to minimize risk for adaptive management using partial or biased knowledge and by testing particular hypotheses[14,53,54], whereas integrative approaches should provide a safer strategy. To overcome these difficulties, our framework offers a promising tool for assessing the level of functional vulnerability in biological communities and guide protection efforts where they are most needed. Moreover, our framework has the major advantage of estimating functional vulnerability by considering reference conditions and even if trait-environment relationships and disturbance regimes are unknown, unpredictable, or poorly documented, a step forward in comparison with previous works[20,39].

Although a few previous works have investigated functional vulnerability[43,55], our framework combines species functional distinctiveness, species abundance distribution and the distribution of functional redundancy across trait space. Most of all, it gives the

advantage to provide an index with absolute values that allows end-users to gauge the intensity of functional vulnerability and to compare sites or periods. Beyond this novelty, our framework is agnostic to how traits mediate specific driver-effect relationships, so we can consider a large range of potential threats that could alter biological communities. Our framework then allows end-users to consider the vulnerability of a system more broadly than when vulnerability is assessed at the species scale or for only one given disturbance[20]. Indeed, considering the lack of knowledge inherent to the variety of pressures and the complexity of biological interactions at the community scale is a necessary step towards efficient ecosystem management[56,57]. Finally, our assessment of functional vulnerability can be applied on any kind of ecosystem or data across space and time (abundance, presence/absence, probability of presence).

Applied to North Sea fishes, we highlight a progressive recovery in demersal communities, which is consistent with previous studies showing the positive ecosystem response to the decrease in demersal destructive fishing (for example, trawling) observed in many European countries[39,58]. Such a congruence between our algorithm predictions and the recovery of the majority of North Sea fish stocks[59], functionally common and distinct species[39], further supports the robustness of our framework, which is based on no preconceived trait-environment relationships. This robustness is also reinforced by finding that functional vulnerability variation due

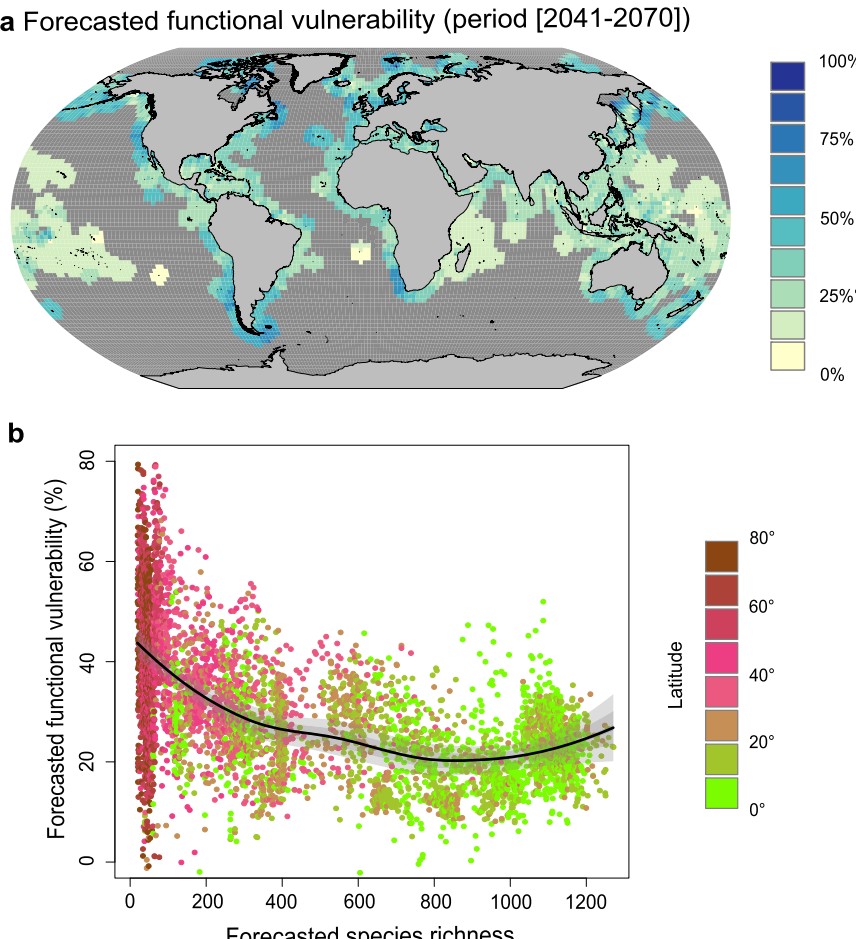

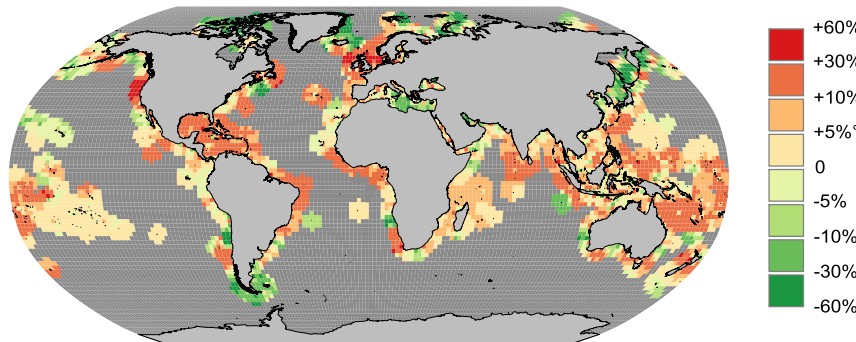

**Fig. 4 | Functional vulnerability of global reef fish communities and associated trends over the coming 60 years. a** Projected functional vulnerability ([2041-2070]) of coastal fish communities under the SSP1-2.6 scenario. **b** Relationship between species richness and functional vulnerability for the period 2041–2070. Data are presented as loess predicted values ± 5(and 10)*standard error. **c** Gain and loss of functional vulnerability between future (2041–2070) and contemporary periods (1981–2015). Source data are provided as a Source Data file. Background map shapefiles are available on the NOAA website: https://www.ngdc.noaa.gov/mgg/shorelines/data/gshhg/latest/.

to simulated random disturbances was three times lower than inter-annual variation. Despite the observed decrease in functional vulnerability over the study period, these results remind us that North Sea fish communities remain particularly vulnerable (90% on average during the last four decades), which is likely explained by the extreme intensity of human pressures in the North Sea during the 20th century[60,61]. Since the end of the 1970s, the exploitation of fish communities in Northeast Atlantic ecosystems has followed the Common Fisheries Policy, mainly characterized by the reduction of

catch quotas and the improvement of gear selectivity[39,62]. These management regulations have therefore contributed to a progressive recovery of fish communities, including functionally common and distinct species in the North Sea[39], which translates into a decrease of functional vulnerability. Even if the North Sea fish communities appear to be slowly recovering, efforts in the ecosystem approach to fisheries management (EAFM) must be maintained, as the absolute value of functional vulnerability remains particularly high. Because of the footprint left by the overfishing era

in this part of the world, the observed decreases in functional vulnerability may not translate into healthier ecosystems.

We find that even in high-diversity systems, such as temperate and tropical regions, the richest communities of marine mammals are among the most vulnerable, supporting the idea that species richness should not be always considered as an umbrella but more likely acting as a buffer against vulnerability[55]. The higher marine mammal vulnerability at northern temperate latitudes was also documented by Albouy et al.[20], who applied a different trait-based approach considering only the effect of climate change. This spatial congruence between the two studies supports the validity of our vulnerability framework, even where trait-environment relationships and disturbance regimes are poorly known. The detection of a positive relationship between species richness and functional vulnerability reinforces the idea that redundancy, and more specifically the distribution of redundancy within a given trait space, is a key element to consider, as the richest communities are not necessarily the most functionally redundant. Conversely, low functional vulnerability was observed in species-poor areas at polar latitudes. So relying on species richness as a proxy for functional redundancy is no longer valid.

Applied on the forecasted reef fish communities, our quantitative framework both revealed a latitudinal gradient in functional vulnerability and how that vulnerability will change in the next decades according to climate change. As expected, species richness appears as a buffer against functional vulnerability (Fig. 4b). However, as for marine mammals (case study 2), a threshold was observed above which the benefit of species richness is counterbalanced by other community characteristics like functional redundancy. The way species traits are distributed within communities can depend on species richness[63] suggesting that the marked functional over-redundancy in species-rich communities may render them more vulnerable to disturbances. It should be noted that the forecasting models that we used only consider fishes' realised environmental niches but ignore biotic habitat availability, which remains a critical step for further development. Additionally, sites where forecasted temperatures were outside of the species' upper thermal niche limit were excluded to prevent any extrapolation. Finally, the selected climate change scenario and horizon in this paper are conservative since it follows SSP1-2.6 scenario by mid-century.

Beyond a simple assessment of functional vulnerability, which is necessary to better plan management actions, our framework also provides the fundamental benefit of better understanding how functional redundancy distribution, abundance distribution, and functional distinctiveness drive the vulnerability of communities. Moreover, despite constructing our index of functional vulnerability to be independent of species richness, in order to prevent trivial effects of species richness, contrasting and highly variable patterns were observed in the relationship between species richness and functional vulnerability. A positive effect of species richness was observed in the North Sea, while a negative relationship was observed for marine mammals. Such a result continues to fuel the diversity-stability debate where the positive effect of species richness is not ubiquitous[64]. Our study is another example showing that functional redundancy, and more specifically its distribution within species trait spaces, may exceed and sometimes counterbalance the effect of species richness, especially when species synchrony is high under random fluctuations, which is the case in most communities[64,65].

Even if our framework provides an integrative assessment of functional vulnerability, meaning that it reflects the vulnerability to a wide panel of potential disturbances, we suggest that future studies could perform similar approaches based on virtual communities differing in functional structure (i.e., functional redundancy, abundances distribution and distinctiveness) where the probability to be impacted differs according to species sensitivity to specific disturbances instead of a random selection of species like we did here. We also highlight

that our approach does not consider the extrinsic vulnerability inherent to the magnitude of environmental changes (i.e. exposure). Indeed, by performing repeated random disturbances on each community (i.e. each site or period), the amount of exposure is constant, so the only factor that drives the level of vulnerability is the species richness, trait and abundance composition within communities. To better define management and conservation strategies in space, the spatial heterogeneity of environmental and anthropogenic disturbances[66,67] needs to be considered in combination with the intrinsic vulnerability requested by decision-makers. Additionally, selected traits should reflect well-defined ecological roles or functions[68] but the lack of data and consensus on the way to choose traits remains a limiting weakness in functional ecology[69,70]. In the worst case where no trait data exist, phylogenetic distances between species can be used as an alternative to compute phylogenetic instead of functional distinctiveness, as proposed by Carmona et al.[71]. We also highlight that our framework could be applied at very local scales when intraspecific trait information is available, all the way through to broad genus or family level traits when trait information is poor. While we assessed functional vulnerability by considering mean trait values for each species, we were unable to account for potential spatio-temporal variation in trait values due to insufficient data. Traits can be highly plastic and can vary in response to environmental (local adaptation) or ecological pressures (phenotypic plasticity) which in turn partly determine the level of functional vulnerability within communities. Finally, we argue that trait-based assessments of ecosystem vulnerability are important to help decision-makers and conservation policy. However, we raise the critical issue that taxonomic biases in trait data availability must be alleviated to avoid further conservation biases orientated towards emblematic and well-studied groups. Nonetheless, trait information is continuously growing[72] for an increasing number of clades, therefore encouraging the confident use of trait-based integrative approaches for conservation perspectives.

The present framework provides only a relative assessment of the functional vulnerability (i.e. relative to all virtual communities constructed from the assessed observed community). The relative approach was chosen to prevent any trivial effect of species richness on vulnerability assessment[71,73]. Indeed, without a relative approach, a species-rich community would need more disturbances to collapse, therefore leading to higher AUC values (from rarefaction curves) and thus lower vulnerability values. The most undesired effect of a non-relative approach would be that some species-poor communities would be trivially identified as the most vulnerable ones while they may not be, and that most management efforts would have to focus on these communities[74]. For example, whilst species-poor temperate assemblages often show higher functional diversity than several richer tropical areas[75], D'agata et al.[76] and Parravicini et al.[55] showed that species-rich systems are not functionally buffered against trait loss. Finally, although the spatio-temporal patterns generated by our framework are congruent with the few previous works that exist, none of these works has proposed an integrated assessment of vulnerability that takes a wide set of potential disturbances into account, therefore preventing an accurate assessment of the robustness relative to vulnerability magnitude.

Preventing ecological impacts of specific disturbance regimes is clearly insufficient, so considering both traits, reference conditions and all potential disturbance regimes appears critical for ecosystem health assessment. By minimizing the error risk in ecosystem health assessment inherent when integrating multiple disturbances, our framework therefore appears as a reliable tool for quantifying the effectiveness of management actions towards more resilient communities and identifying areas where conservation is and will be needed. Concretely, our framework provides an integrative estimate of ecosystem functional vulnerability that could provide key insight for achieving some Sustainable Development

Goals targets[77]. The present framework will provide a critical tool for decision-makers to help identify management priorities based on the inherent vulnerability of ecosystem functions to disturbances and better guide the expansion of the current network of protected areas towards the next target of 30% coverage by 2030[78]. Additionally, our framework can contribute to better identify which properties determine vulnerability to the myriad of threats biological communities will face in an uncertain future.

## Methods

### Virtual communities framework
The first step of the framework is to generate a multi-dimensional trait space that represents the functional similarity among all species[79–81], to characterize the functional structure of a given community. The second step is to compute a distance matrix that quantifies the dissimilarity between all pairs of species based on their trait combinations. Here, we used the flexible Gower pairwise distance since it can handle multiple data types (from continuous to categorical traits) and is less sensitive to missing data than other distances[81]. We used the *compute_dist_matrix()* function in the *funrar* R package[82]. Finally, we performed a Principal Coordinates Analysis using the *pcoa()* function (*ape* R package) to transform pairwise species distances into a multidimensional space[83]. From this trait space of the observed community, we generated virtual communities by creating scenarios crossing three community properties: (i) distribution of functional redundancy in the trait space, (ii) distribution of abundance across species and iii) relationship between species trait distinctiveness and species abundance (Fig. 1). These virtual communities represented a panel of theoretical extremes of trait configurations within communities. For each of the three community properties, we used several modalities to characterize extreme cases.

For functional redundancy, we used three modalities: homogeneous, heterogeneous and the distribution of the observed community (Fig. 1). We achieved homogeneous and heterogeneous distributions by moving species within the trait space. To do so, we placed a $20 \times 20$ regular grid cell on the observed two-dimensional trait space (note that functional entities can be defined as multidimensional cells when the trait space has more dimensions), the limits of the grid being defined by the upper, lower, leftmost and rightmost species in the trait space. We defined each cell as a 'functional entity' since the position in the trait space is a proxy of species roles in ecosystems. We recommend considering only the two first PCoA axes as considering more axes would scatter species into a very high number of functional entities (i.e. grid resolution^number of selected PCoA axes) resulting in many more cells with only few or only one species in them. In extreme cases, every single functional entity would only hold one species, reducing the interest of the framework, since it is not likely to have so many functions and so few species per cell/functional entity/function. For homogeneous redundancy, the number of species per functional entity corresponds to the rounded (down) value of the ratio between the total number of species in the community (i.e., species richness S) and the number of functional entities (N) for which at least one species is present. For heterogeneous redundancy, we placed only one species within each functional entity and the rest of the species (S minus N) were all placed in the last functional entity having over-redundancy[43].

We used three modalities to represent the distribution of abundances across species: homogeneous, heterogeneous, and the observed distribution of abundances between species (Fig. 1). We obtained the homogeneous distribution by attributing the same abundance to all species. We obtained the heterogeneous distribution of abundance by attributing only one individual to each species, except for the last one to which the rest of the total abundance was attributed. The total number of individuals in the virtual communities always equaled the total abundance of the observed community.

To infer the relationship between species trait distinctiveness and abundance, we selected two modalities: negative and positive (Fig. 1). Trait distinctiveness $D_i$ is an index quantifying how functionally dissimilar, on average, a given species is compared to all other species in the regional pool[42,82]

$$D_i = \frac{\sum_{j=1, j \neq i}^{S} d_{ij}}{S - 1} \tag{1}$$

where $S$ is the total number of species within the species pool, and $d_{ij}$ is the dissimilarity between species $i$ and $j$. In this index trait dissimilarity was standardized and consequently the distinctiveness ranged between 0 and 1. A functionally distinct species, i.e., with a high value of $D_i$, corresponds to a species with original trait values compared to the rest of the regional species pool. Distinctiveness values were computed using the *distinctiveness()* function of the *funrar* R package. To obtain negative or positive relationships between $D_i$ and species abundance, we ordered $D_i$ values and species abundances to attribute the highest abundances to the least or the most functionally distinct species, respectively.

For a given community that we aim to quantify the functional vulnerability, we simulated a series of disturbances on each of its associated virtual communities and on the observed community itself. At each disturbance, we applied a decrease in the abundance of randomly selected species (we recommend to apply a decrease of 5% of the total abundance of the community, the selected number of impacted species only impacting time calculation) and we recalculated new abundances and then the number of functional entities (i.e., grid cells) with at least one species still present. These simulated disturbances were applied successively until there were no remaining species in any functional entities (see rarefaction curves, Fig. 1b). Finally, the functional vulnerability of a community varied between 0 and 100% and was computed as follows:

$$Functional\ vulnerability = 100 \times \frac{AUC_{obs} - AUC_{min}}{AUC_{max} - AUC_{min}} \tag{2}$$

With $AUC_{max}$ being the area under the rarefaction curve relative to the least vulnerable virtual community (Fig. 1b), $AUC_{min}$ the area under the curve (AUC) relative to the most vulnerable virtual community (Fig. 1b), and $AUC_{obs}$ the area under the curve relative to the observed community (Fig. 1b). A community for which functional vulnerability is equal to 0% will be the community that will lose its functions the slowest among all virtual communities (i.e., when the curve of the observed real community is superimposed to the one of the least vulnerable virtual community). By contrast, a community with functional vulnerability equal to 100% can be considered as the most vulnerable virtual community (i.e., when the curve of the observed real community is superimposed to the one of the most vulnerable virtual community). The entire statistical procedure can be repeated several times to prevent any overinterpretation (see the 'Sensitivity analyses' section).

### Adaptation of the algorithm to occurrence and habitat suitability data
Because presence/absence information is more accessible in ecological surveys, we developed a second algorithm to assess functional vulnerability when species abundance data are not available. While the statistical procedure was the same as the one developed in the abundance-based approach, the rarefaction curve of the observed community is here compared to the rarefaction curves of only two virtual communities characterized by opposite functional redundancy distributions (Supplementary Fig. 1) and species extinction were used as disturbance. To account for datasets containing probabilities of presence (or habitat suitability) usually computed from presence/

absence data, we developed a third version of the algorithm. Disturbances are here characterized by a diminution of 0.05 on each value of habitat suitability (i.e. 5% of the maximum value 1). We applied three case studies demonstrating the application of our algorithms to these different forms of biodiversity data (i.e., abundance estimates, range maps, and habitat suitability models).

## Case study 1: Temporal vulnerability of North Sea fish communities

We applied our quantitative framework to estimate the functional vulnerability of the North Sea fish communities. We gathered abundance data from the International Bottom Trawl Survey (IBTS), an annual monitoring campaign that uses stratified random sampling to survey fish and invertebrate communities in 1° longitude × 0.5° latitude cells covering the entire North Sea each year in winter (Fig. 2a). Every year, at least two hauls were performed within each cell with a 3-m vertical opening bottom trawl (GOV trawl) with a 10-mm mesh codend during 30 min at an average speed of 4 knots. In each survey, fishes were identified and counted, and the resulting abundances were standardized to numbers of individuals per hour of trawling. The fish abundance data set included 168 taxa across 154 survey cells over 38 years (1983-2020), which represents a total of 13,433 hauls. Prior to statistical analyses, species abundances (across all survey cells and years) were $\log10(x+1)$ transformed. We calculated the yearly mean abundance of each species at the scale of the North Sea to obtain one regional community per year from 1983 to 2020. Abundance data were downloaded from the International Council for Exploitation of the Sea data portal (ICES; https://datras.ices.dk/Data_products/Download/Download_Data_public.aspx). For this case study, each simulated disturbance corresponded to a decrease of 5% of the total abundance in the observed community.

In combination with species abundance, and for each fish taxa, eight traits encompassing life history, habitat use and trophic ecology—chosen because they are documented to mediate species influences on ecosystem functioning[39,84]—were collected from the PANGAEA traits database:[85] age at maturity, asymptotic length, fecundity, offspring size, habitat (water column position), trophic level, feeding mode, and spawning type. We applied the abundance-based version of the algorithm to estimate the functional vulnerability of the North Sea fish community per year (https://figshare.com/s/9d3cd1d6f68a73dcea11).

## Case study 2: Global distribution of marine mammals vulnerability

Marine mammals range maps for 122 species were downloaded from the IUCN website (http://www.iucnredlist.org). From these range maps, we computed a presence/absence matrix by overlapping the species' range maps and counted the number of species that occured in each 1° grid cell[20]. We collected fourteen traits covering five main functions (feeding, habitat, reproduction, social behavior) usually included in climate change vulnerability assessments (i.e., diet, foraging depth range, foraging fasting strategy, terrestriality, female sexual maturity, weaning, gestation length, inter-litter interval, social group size, adult mass body mass, sexual dimorphism, breeding location, social behavior; see ref. 20 for further details). For this case study, we applied the presence–absence version of our algorithm to estimate the functional vulnerability of each marine mammal community at global scale (https://figshare.com/s/5dcefa52f529ad34dc02).

## Case study 3: Future changes in functional vulnerability of reef fishes

In this last study case, we applied our algorithm to estimate how climate change could affect the functional vulnerability of global reef fish communities in 60 years (2041–2070 compared to the period of reference 1981–2015). To do so, we estimated the functional vulnerability of contemporary and future communities using species

distribution models of 2320 species. The species distribution modeling framework is detailed in the supplementary information. In brief, we integrated > 12 million records of species presence from open access databases (GBIF[45], OBIS[46]) and local SCUBA transects (Reef Life Survey[47,48], SERF[49], GASPAR[50]) (Supplementary Fig. 12). We extracted the environmental conditions of species' presences for 6 environmental variables: sea surface temperature (SST; minimum and maximum), pH (minimum), sea surface salinity (SSS; minimum), net primary productivity (NPP; mean), degree heating weeks (mean; DHW[86]) and gravity of human impacts within 500 km² (mean) and used these variables as covariates. Although we used three metrics containing information on sea surface temperature, SST-min and SST-max were not correlated with degree heating weeks which is a measure of heat stress rather than absolute temperature. We used generalized linear models, generalized additive models, and random forests—algorithms that cover a range of complexity relating the response variable to covariates—to explain the relationship between species presence and the local environmental conditions (see Supplementary Methods). For each model, we checked that the Pearson correlation between pairs of variables was <0.7, and if not, retained the variable with the highest deviance explained and lowest error rate (depending on the model algorithm, see below). We iterated this process until we obtained a set of uncorrelated and maximally explanatory variables. We used a target-group pseudo-absence approach[87], whereby we generate 5-folds of background data which reduces model biases occurring in modeling presence-only data. We performed model evaluations using spatially blocked 5-fold out-of-sample predictions retaining only models with a true-skill statistic > 0.35[88]. Our models were well-calibrated having a mean sensitivity of 0.89 ± 0.06, specificity of 0.75 ± 0.09, TSS of 0.64 ± 0.12, and an AUC statistic of 0.84 ± 0.07 (Supplementary Fig. 13). Having well-performing models for 2320 species, we next predicted the habitat suitability of species in present-day environmental conditions and under environmental conditions for 2041–2070 under SSP1-2.6 climate change scenario (making separate suitability predictions based on projection outputs of seven CMIP6, Coupled Model Intercomparison Project phase 6, Earth system models[89]). We then assembled (mean-averaged) all predictions across all model algorithms, CMIP6 models and background data iterations to avoid spurious model outputs from these sources of uncertainty. We presented the SSP1-2.6 climate change scenario (Shared Socioeconomic Pathways 1 combined with the Representative Concentration Pathway 2.6[23]) as it is inherent to the Paris Agreement that aims to limit global warming to less than 2 °C[51]. We modified the SST, SSS, pH, and NPP under future conditions but retained all other variables at their current day values. We only included DHW as a static variable for multiple reasons: (i) primarily to capture the effects of past thermal disturbances on current distributions, (ii) most locations globally will have degree heating weeks well beyond current values, leading to a high level of extrapolation in predicting into future scenarios[90] (oppose to temperature which is only extrapolated in the tropics). Note that DHW had by far the lowest explanatory power of all variables on average across all species such that our results are likely to be robust to excluding degree heating week projections (if temperature change is already included as SST-min and SST-max). Finally, dispersal rates of species were constrained to 10 km per year, corresponding approximately to observed realized latitudinal range shifts of marine species[91]. All species maps were produced on a consistent 0.25° global grid that represents coastal and reef systems.

For this last case study, 12 traits were chosen: length at maturity, age at maturity, maximum age, the asymptotic mass and length, the Von Bertalanffy growth coefficient (from FISHLIFE[52]), mobility, period of activity, schooling, vertical position, diet, and reef association (from the GASPAR traits database[50]). We applied the suitability-based version of the algorithm to estimate the functional vulnerability (https://figshare.com/s/2c5fb050f2d2b9e01170).

## Sensitivity analyses

To test the potential influence of the number of disturbances series on AUC value and, therefore, on the functional vulnerability index, we performed sensitivity analyses (Supplementary Fig. 2). Applied to the North Sea fish community (case study 1), this analysis revealed that the relative standard deviation of AUC values remained particularly low (always less than 2% by median), regardless of the number of disturbances. For a good compromise between computation time and robustness, we applied 2 series of 500 disturbances on each community, which corresponds to 1.8% in uncertainty (see Supplementary Fig. 2). A sensitivity analysis was also carried out to test the influence of the grid resolution applied to the species traits space. A 20 × 20 grid cell resolution was the minimal resolution needed to get the most representative values of functional vulnerability (Supplementary Fig. 3).

Several analyses were performed to quantify the effect of missing trait data (i.e., Not Available, or NA in traits datasets) on final vulnerability values. To do so, the spatio-temporal distribution of NAs was firstly investigated and vulnerability values were recomputed by removing all species for which a NA was observed for at least one trait. We then compared the new vulnerability values to those obtained from the entire trait dataset (i.e., with all NAs). Additionally, we re-ran the analyses considering implemented traits tables (i.e., with all NAs replaced by estimated values), which were obtained using a random Forest technique (missForest: *MissForest R package*; Stekhoven and Bürhlmann[92]). For all case studies, vulnerability values presented a very low sensitivity to the NA items present in traits datasets, as their deletion and/or imputation did not substantially affect spatio-temporal patterns of vulnerability (Supplementary Figs. 4–9).

To investigate the effect of trait deletion on vulnerability patterns, the algorithm was applied for several trait combinations and by varying the total number of selected traits. For each number of selected traits, we performed 20 iterations (i.e. 20 combinations of randomly selected traits). We then compared final outputs to our reference values coming from the analysis based on all traits. For all case studies, vulnerability values present a very low sensitivity to trait deletion, as spatio-temporal patterns of the vulnerability were similar to those obtained from analyses using all traits (Supplementary Fig. 10).

All statistical analyses were performed under the R environment (version 4.1.0; R Core Team, 2021).

## Reporting summary

Further information on research design is available in the Nature Research Reporting Summary linked to this article.

## Data availability

Abundance data of the North Sea fish communities (case study1) are available through https://datras.ices.dk/Data_products/Download/Download_Data_public.aspx and at https://figshare.com/s/9d3cd-1d6f68a73dcea11 (input table for analyses). Traits data of the North Sea fish communities are available through the PANGAEA traits database at: https://doi.org/10.1594/PANGAEA.900866. Presence/absence and traits data for marine mammals (case study 2) are available at: https://figshare.com/articles/Input_data_for_Global_vulnerability_of_marine_mammals_to_global_warming_/11323304. All fish species occurrences used for SDM models (GASPAR, SERF, RLS, OBIS & OBIS) and associated forecasted habitat suitability data (case study 3) are available at [https://figshare.com/s/2c5fb050f2d2b9e01170]. Traits data used for case study 3 (GASPAR & FISHLIFE) are available at: [https://figshare.com/s/2c5fb050f2d2b9e01170]. Projection of environmental variables from the CMIP6 Earth system model are available through https://esgf-node.llnl.gov/projects/cmip6/. Source data are provided with this paper: [https://doi.org/10.6084/m9.figshare.20099219]. Background map of Fig. 2a was obtained through the *maps* r package (v.3.3.0). Background map shapefiles for Figs. 3a, b, 4a,

c are available on the NOAA website: https://www.ngdc.noaa.gov/mgg/shorelines/data/gshhg/latest/.

## Code availability

Codes to run vulnerability analyses are available at:
Case study 1: https://figshare.com/s/9d3cd1d6f68a73dcea11
Case study 2: https://figshare.com/s/5dcefa52f529ad34dc02
Case study 3: https://figshare.com/s/2c5fb050f2d2b9e01170
The R code used to generate figures is available in the figshare repository: [https://doi.org/10.6084/m9.figshare.20099219].

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

## Acknowledgements

This research is supported by the Fondation pour la Recherche sur la Biodiversité (FRB) and Electricité de France (EDF) in the context of the CESAB project 'Causes and consequences of functional rarity from local to global scales' (FREE) and by FRB and FFP (France Filière Pêche) in the context of the CESAB project 'Climate change effects on exploited marine communities' (MAESTRO). This research was also funded through the 2017–2018 Belmont Forum and BiodivERsA REEF-FUTURES project under the BiodivScen ERA-Net COFUND program along with the French National Research Agency, the Natural Sciences and Engineering Research Council (Grant No. RGPBB/525590), the Canada Research Chairs Program, the Ocean Frontier Institute, and the Research Council of Norway (no. 295340). AT was supported by the Estonian Ministry of Education and Research (PSG505). We acknowledge Charlie Gough (from Blue Venture: https://blueventures.org/about), Ivor Williams (from the NOAA Coral Reef Ecosystems Division in Hawaii), Michel Kulbicki (from the GASPAR project), Nicholas A.J. Graham, Andrew Hoey, David Booth, Alan Friedlander, Shaun Wilson, Pascale Chabanet, Jessica Zamborain Mason, Eran Brokovich, Marah Hardt, Sebastian Ferse, Joshua Cinner, Laurent Wantiez, the Gouvernement de la Nouvelle-Calédonie, province Sud de la Nouvelle-Calédonie and province Nord de la Nouvelle-Calédonie for SERF data provision. We thank all the people who participated in the fieldwork. We acknowledge Rick Stuart-Smith and Graham Edgar (University of Tasmania) for their help on RLS data. We finally acknowledge the AquaMaps team and Cristina Garilao for data provision.

## Author contributions

A.A. drafted the manuscript. A.A., C.W., C.A., and A.M. led the analyses. A.A., C.W., A.M., E.G., C.A., A.C.A., M.M., A.B., A.L.G., M.T., L.V., K.K., K.K.R., M.B., J.T., A.T., C.V., N.M., W.T., and D.D. contributed to the interpretation of the data and drafted the manuscript.

## Competing interests

The authors declare no competing interests.

## Additional information

[1]IFREMER, Unité Halieutique Manche Mer du Nord, Laboratoire Ressources Halieutiques, Boulogne-sur-Mer, France. [2]Division of Aquatic Ecology and Evolution, Institute of Ecology and Evolution, University of Bern, Bern, Switzerland. [3]Department of Fish Ecology and Evolution, Center for Ecology, Evolution and Biogeochemistry, Eawag - Swiss Federal Institute of Aquatic Science and Technology, Kastanienbaum, Switzerland. [4]EDF R&D LNHE - Laboratoire National d'Hydraulique et Environnement, 6 quai Watier, Chatou, France. [5]Unité Biologie des Organismes et Ecosystèmes Aquatiques (BOREA), Muséum National d'Histoire Naturelle, Sorbonne Université, Université de Caen Normandie, Université des Antilles, CNRS, IRD, Paris, Cedex 05, France. [6]Ecosystems and Landscape evolution, Institute of Terrestrial Ecosystems, Department of Environmental Systems Science, ETH Zürich, Zürich, Switzerland. [7]Unit of Land Change Science, Swiss Federal Research Institute WSL, Birmensdorf, Switzerland. [8]Department of Biology, Lakehead University, Thunder Bay, ON, Canada. [9]Department of Biology, Dalhousie University, Halifax, NS, Canada. [10]IFREMER, unité Ecologie et Modèles pour l'Halieutique, rue de l'Ile d'Yeu, BP21105, Nantes, cedex 3, France. [11]Red Sea Research Center, King Abdullah University of Science and Technology, Thuwal, Saudi Arabia. [12]Institute of Marine Science, University of Portsmouth, Ferry Reach, Portsmouth, UK. [13]CGG, Crompton Way, Crawley, UK. [14]UMR ENTROPIE, IRD-UR-UNC-IFREMER-CNRS, Centre IRD de Nouméa, Nouméa Cedex, New-Caledonia, France. [15]Department of Biometry and Environmental Systems Analysis, Albert-Ludwigs-University of Freiburg, Freiburg, Germany. [16]Quantitative Aquatics, G.S. Khush Hall, IRRI, Los Baños, Philippines. [17]School of Biology, Faculty of Biological Sciences, University of Leeds, Leeds, UK. [18]Centre for Biodiversity and Conservation Science, School of Biological Sciences, University of Queensland, Brisbane, Australia. [19]NORCE Norwegian Research Centre, Bjerknes Centre for Climate Research, Bergen, Norway. [20]Institute of Ecology and Earth Sciences, University of Tartu, Tartu, Estonia. [21]CEFE, Univ Montpellier, CNRS, EPHE, IRD, Montpellier, France. [22]CESAB – FRB, Montpellier, France. [23]UMR MARBEC, Univ Montpellier, CNRS, IFREMER, IRD, Montpellier, Cedex, France. [24]Univ. Grenoble Alpes, Univ. Savoie Mont Blanc, CNRS, LECA, Laboratoire d'Ecologie Alpine, Grenoble, France. [25]Institut Universitaire de France, Paris, France. ✉e-mail: Arnaud.Auber@ifremer.fr

