## [Peer Review File · Nature Communications]

A functional vulnerability framework for biodiversity conservationREVIEWER COMMENTS

Reviewer #1 (Remarks to the Author):

The manuscript from Auber et al. presents and applies a new approach to conceptualize and quantify functional vulnerability of ecosystems. Predicting how the ecological roles or functions in an ecosystem might be affected by future changes in species' abundance or extinctions presents a conceptual and analytical challenge that is important to overcome as it can identify where conservation should be prioritized. The authors' new approach 1) examines how coverage of functional space (entities) of observed community responds to successive extinctions (disturbances), and 2) compares this response to 'null' communities created by redistributing species in trait space and abundance across species (Fig. 1 in the manuscript), to 3) compute a new functional vulnerability metric capturing how rapidly functional space is lost relative to the null communities. Application of the framework to three different marine case studies is impressive and aids to showcase the potential of this new approach, and provides valuable insights into these important cases. Overall, I think this work has great potential for helping ecologists, managers, and policy makers to understand the functional vulnerability of ecosystems in a comparable way. Nevertheless, I have some queries about the manuscript that I will detail below.

I am not convinced about the framing of the manuscript. From the title and the Introduction I was expecting the precautionary approach, multiple stressors, and unpredictability to be directly incorporated into the metric and the case studies. However, more simply, the metric applies random disturbances to the community and examines how this changes occupation of the trait space relative to a full range of contrasting scenarios. While I really value the advance made here, I feel that too much spin is applied to the random scenario of species disturbances – these kinds of scenarios have been applied in many other biodiversity studies; I do not see how this is 'precautionary', especially since functional vulnerability is based on the randomly applied disturbance scenario in the observed community, rather than a scenario where e.g. the most abundant or most functionally unique species are lost first.

Furthermore, the Discussion paints a uniformed approach to application of disturbances across species (i.e., random) as a real advantage of this approach, but I find this over-sold: is it really an advantage, or a necessary simplification and limitation? Does it really 'account for the lack of knowledge' or just accept and work with a lack of knowledge? To my knowledge, some of the most closely allied work to this was done by members of this team on coral reef fish. The authors may wish to explicitly address in the manuscript how this work advances over previous efforts to look at metrics such as functional vulnerability and over-redundancy?

I would like the authors to consider and discuss more what the metric means. The interpretation of the metric all depends on the 'null' communities: these are constrained by the number of species, total abundance, and the size of the functional space. My interpretation is that it's saying: "how functionally vulnerable is this community relative to a worse-case-scenario given the same number of species, individuals (or biomass), and functional richness"? So, the functional vulnerability metric is always relative to a set of community-specific parameters. How reasonable and comparable is that? Do we expect this to accurately identify areas that will lose more function with species loss (what I would interpret as functional vulnerability), or just areas that are closer to some worse-case scenario given their community parameters (which may constrain their vulnerability in absolute terms relative to other communities)? This has important implications for the meaning of functional vulnerability.

Some aspects of the methods need more detail: 1) It is not clear how abundance reductions are applied. Do the authors take random draws from a normal distribution and apply these reductions to some kind of normalised relative abundances? 2) How does this method deal with highly dimensional functional spaces? How do the authors

divide up highly dimensional functional spaces into functional entities?

Reviewer #2 (Remarks to the Author):

This paper explore the performance of a trait-based framework to assess the vulnerability of biological communities to extinction. For that, they used three working examples of marine communities using past data of species abundances, current presence absence data, and future projections due to climate change. The results show that the trait-based framework, which involves the comparison of observational data with simulated data, is able to document differences in functional vulnerability across the three datasets, although there is an inherent context dependency at both temporal and spatial scales.

I think the aim of this paper is interesting, and goes in line with the current line of thinking that we do not need to know every single mechanistic aspect of the relationship between species characteristics and their vulnerability to predict their responses to environmental changes or perturbations. Moreover, the method has all the ingredients to be adopted by a wide range of ecologist and conservation biologists because it is tractable (the algorithms are well defined), comparable (outcomes can be compared across space and time) and scalable (it can be applied to a wide range of scales).

Yet, I am confused by one important aspect. The framework is about predicting based on precaution principles. It is about helping decision makers where to put the focus on given time and monetary limitations before the communities are more degraded. However the paper does not present any analysis of the accuracy of the algorithms. It just presents differences in vulnerability across communities, but there is no validation of at least some of these results. In such respect, the authors state in lines 275-277 that the results of the trait framework here presented is congruent with previous studies. In general lines I agree with some of the results presented. Communities in the Northern Hemisphere are more vulnerable compared to other regions (Fig.3), but there are other regions in the world that they are also highly vulnerable (South China sea). Similarly, it seems that the vulnerability based on trait information of the Northern sea has slightly decreased, but to what extent is this also true. In sum, my worry here is that there is not info provided to know to what extent the vulnerability to extinction based on trait info is over or underestimated. The overall temporal or spatial patterns across dataset makes sense but the magnitude is what it remains unclear.

My second point is that the method does not distinguish between inherent vulnerability to extinction due to the species characteristics and their abundance from extrinsic vulnerability due to the combination of several perturbations. I can imagine being a decision maker and asking whether it will be effective to focus on a particular community or not. It might be the case that even if I put all the effort in diminishing the impacts and perturbations occurring of a given community its intrinsic vulnerability will be still too high, and therefore, it would make more sense to focus on another communities in which my management could be more effective. I think this point needs to be addressed in order to engage managers and decision-makers. Otherwise, the paper remains a nice but just modeling exercise.

Finally, It is not clear from reading the methods how complete is the trait data. I assume there was no information for all traits for all species. how then the authors proceeded? Did they impute trait data? did they considered NAs? I think this is an important aspect, which leads me more to more general question. How sensitive is the trait framework presented here to have several biases of trait information? I can imaging four sources of biases. 1) Low number of traits, 2) trait information available for some particular communities, 3) trait information available for some particular taxa (families), and trait information obtained for different species across different years. Since the key aim of

the paper is based on applying trait info to predictive tools, this must be addressed in the same way that the authors have considered a sensibility analyses based on the number of perturbations.

As a minor comment, I acknowledge that the paper can not tackle with every single aspect, but I think it worth mentioning in the discussion that the traits considered here do not change across time or species, so the paper ignores the potential effect of trait variation, both genetic (local adaptation) or ecological (phenotypic plasticity) on driving vulnerability differences across communities. If the discussion includes the importance of trait variation within populations, I think is it also worth including the potential of trait variation for the same species across time and space.

REVIEWER COMMENTS

Reviewer #1 (Remarks to the Author):

The manuscript from Auber et al. presents and applies a new approach to conceptualize and quantify functional vulnerability of ecosystems. Predicting how the ecological roles or functions in an ecosystem might be affected by future changes in species' abundance or extinctions presents a conceptual and analytical challenge that is important to overcome as it can identify where conservation should be prioritized. The authors' new approach 1) examines how coverage of functional space (entities) of observed community responds to successive extinctions (disturbances), and 2) compares this response to 'null' communities created by redistributing species in trait space and abundance across species (Fig. 1 in the manuscript), to 3) compute a new functional vulnerability metric capturing how rapidly functional space is lost relative to the null communities. Application of the framework to three different marine case studies is impressive and aids to showcase the potential of this new approach, and provides valuable insights into these important cases. Overall, I think this work has great potential for helping ecologists, managers, and policy makers to understand the functional vulnerability of ecosystems in a comparable way. Nevertheless, I have some queries about the manuscript that I will detail below.

We thank the reviewer for this positive evaluation of our manuscript and thus addressed all your queries in the revised version. As general information, we recomputed vulnerability values both for case studies 1 and 2 because minor mistakes were detected in abundance/occurrence tables from the previous manuscript version, but these modifications did not affect the final patterns.

For the North Sea, a *round()* function was applied on abundances values (number of individuals per hour of trawling) prior to vulnerability calculation in the previous version. As a consequence was that the abundance of very rare species (<1 ind.km²) was then converted to zero, so they were considered absent from the communities. We thus re-run all analyses (including all sensitivity analyses of the supplementary material) without the *round()* transformation. We did not observe any major change.

For Marine mammals, we re-ran all analyses because one species (out of 122) was lacking in the previous version and used the same traits combinations as Albouy *et al.*, 2020. We did not observe any consistent change.

I am not convinced about the framing of the manuscript. From the title and the Introduction I was expecting the precautionary approach, multiple stressors, and unpredictability to be directly incorporated into the metric and the case studies. However, more simply, the metric applies random disturbances to the community and examines how this changes occupation of the trait space relative to a full range of contrasting scenarios. While I really value the advance made here, I feel that too much spin is applied to the random scenario of species disturbances – these kinds of scenarios have been applied in many other biodiversity studies; I do not see how this is 'precautionary', especially since functional vulnerability is based on the randomly applied disturbance scenario in the observed community, rather than a scenario where e.g. the most abundant or most functionally unique species are lost first.

We agree that a clarification was needed. We see the point of the reviewer: considering the worst scenario is precautionary. Yet, it would imply to consider only one scenario with clear expectations or hypotheses while uncertainty is central to the concept of precautionary principle since its application promotes reflection in the face of uncertainty, arguably leading to better strategies.

We understand the point of the reviewer that single-scenario approaches (for example, where the most distinct species are removed first) can be useful in implementing a precautionary approach. However, single and multiple approaches provide different information and both can contribute to implement a precautionary approach. With reference to Kriebel *et al.*, (2001), the precautionary principle is characterized by 4 components:

1. Taking preventative action in the face of uncertainty
2. Shifting the burden of proof to the proponents of an activity
3. Exploring a wide range of alternatives to possible harmful action
4. Increasing public participation in decision making

With respect to component 1, we think that our approach fits well, and can advance the precautionary approach, in that it provides information in the 'face of uncertainty'. We have limited ability to predict precisely what

future impacts will be, how they will interact, or how populations/species will respond, especially in complex natural communities. Thus our approach provides estimates of vulnerability in the face of a lack of detailed scientific information on what impacts will be and how species will respond. Thus we think it provides valuable information that is not captured in single scenario approaches - such approaches fail to consider the uncertainty in future outcomes and thus can't provide the same information as our approach.

The concrete repercussion of the precautionary principle in the assessment of the integrated vulnerability is given by the fact that the metric considers a wide set of disturbances through the application of abundance reduction on randomly selected species. Applying repeated random disturbances directly embraces uncertainty and accounts for the potential that anything could happen. In this way, the metric provides a good overview of the 'mean' response to all potential disturbances that could affect communities, therefore providing a precautionary estimation of the vulnerability. In all cases we agree with the reviewer that the precautionary principle is not strictly incorporated in the metric on a mathematical aspect. We therefore clarified this point in the revised version, notably through this sentence in the abstract :

"Here, we present a novel functional vulnerability framework that quantitatively incorporate uncertainty and ecological traits into a generalizable tool. By quantifying the vulnerability of communities to a wide range of threats (where driver-response relationships are unknown) we assist in applying the precautionary principle to biodiversity conservation."

Clarifications (**important : please use the mode 'final version' for line numbers, not use the 'final with tracked changes' view**) are now included from l. 54-58 & 121-129.

Related references:

Kriebel, David, Joel Tickner, Paul Epstein, John Lemons, and Richard Levins. 2001. "The Precautionary Principle in Environmental Science." *Environmental Health Perspectives* 109 (9).

Furthermore, the Discussion paints a uniformed approach to application of disturbances across species (i.e., random) as a real advantage of this approach, but I find this over-sold: is it really an advantage, or a necessary simplification and limitation? Does it really 'account for the lack of knowledge' or just accept and work with a lack of knowledge? To my knowledge, some of the most closely allied work to this was done by members of this team on coral reef fish. The authors may wish to explicitly address in the manuscript how this work advances over previous efforts to look at metrics such as functional vulnerability and over-redundancy?

The unified approach can be considered as an advantage rather than a necessary simplification. In the case where we aim to assess the effect of a specific or well-known single disturbance our approach should not be applied because we could have expectations about which species will get lost first. This is rarely the case in Nature, so the framework proposed in our study should be applied when we aim to assess the vulnerability to all potential types of disturbance that could occur. So, our approach can be seen as an advantage since it provides a global/integrated assessment of functional vulnerability, which is of particular interest to stakeholders and ecosystem managers in face of uncertainty. Our framework does not exclude that other scenarios can be tested or simulated as new knowledge arises. We now point out this complementarity lines l. 54-58 & 121-129. We also reworded 'account for the lack of knowledge' by 'considering the lack of knowledge' which is more appropriate (l. 299).

Contrary to previous works that also aimed to quantify the functional vulnerability, our framework embraces the uncertainty associated to species response and future threats by adopting a multiple-scenarios approach. Moreover, our framework is more integrative than previous works by considering both functional rarity, redundancy and species abundances. To further stress that point, we included new sentences in the revised manuscript (l. 128-131).

I would like the authors to consider and discuss more what the metric means. The interpretation of the metric all depends on the 'null' communities: these are constrained by the number of species, total abundance, and the size of the functional space. My interpretation is that it's saying: "how functionally vulnerable is this community relative to a worse-case-scenario given the same number of species, individuals (or biomass), and functional richness"? So, the functional vulnerability metric is always relative to a set of community-specific parameters. How reasonable and comparable is that? Do we expect this to accurately identify areas that will lose more

function with species loss (what I would interpret as functional vulnerability), or just areas that are closer to some worse-case scenario given their community parameters (which may constrain their vulnerability in absolute terms relative to other communities)? This has important implications for the meaning of functional vulnerability.

First, we reiterate that our approach is not considering only the worst-case scenario but instead is based on uncertainty to embrace multiple scenarios. Otherwise, we completely agree with your comment that the approach developed here is relative, i.e. influenced by a set of community-specific parameters. and more information must be added to fully understand the metric. By considering species richness as one of these parameters, we adopted the relative approach to prevent any trivial effect of species richness on vulnerability assessment (Carmona *et al.*, 2017; de Bello *et al.*, 2016; Mouillot *et al.* 2014). Indeed, in an absolute version of the approach, a species-rich community would need a more severe disturbance to collapse, therefore leading to a higher AUC value (from rarefaction curves) and thus a lower vulnerability value. The most undesired consequence of an absolute approach would be that many species-poor sites would be trivially identified as the most vulnerable ones while they may not be, and that most management efforts should focus on these sites (Boyer & Jetz 2014). For example, it has already been reported that species-poor temperate assemblages show higher functional diversity than several richer tropical areas (Stuart Smith *et al.*, 2013) while D'agata *et al.* (2014) and Parravicini *et al.* (2014) show that even species-rich systems are not functionally buffered against trait loss. We included new sentences to specify and justify the choice of the relative approach (see l. 394-404).

Related references also included in the revised version:

- Carmona, C.P., Guerrero, I., Morales, M.B., Oñate, J.J. & Peco, B. (2017) Assessing vulnerability of functional diversity to species loss: a case study in Mediterranean agricultural systems. *Functional Ecology*, 31, 427–435.
- de Bello, F., Carmona, C.P., Leps, J., Szava-Kovats, R. & Pärtel, M. (2016) Functional diversity through the mean trait dissimilarity: resolving shortcomings with existing paradigms and algorithms. *Oecologia*, 180, 933 – 940.
- Mouillot, D., Villéger, S., Parravicini, V., Kulbicki, M., Arias-Gonzalez, J. E., Bender, M., Chabanet, P., Floeter, S. R., Friedlander, A., Vigliola, L. & Bellwood, D. R. Functional over-redundancy and high functional vulnerability in global fish faunas on tropical reefs. *Proc. Nat. Acad. Sci*, **111**, 13757–13762 (2014).
- Boyer, A.G. & Jetz, W. (2014). Extinctions and the loss of ecological function in island bird communities. *Glob. Ecol. Biogeogr.*, 23, 679 – 688.
- Parravicini, V., Villéger, S., McClanahan, T.R., Arias-González, J.E., Bellwood, D.R., Belmaker, J., Chabanet, P., Floeter, S.R., Friedlander, A.M., Guilhaumon, F., Vigliola, L., Kulbicki, M. & Mouillot, D. (2014) Global mismatch between species richness and vulnerability of reef fish assemblages. *Ecology Letters*, 17, 1101–1110.
- Stuart-Smith, R.D., Bates, A.E., Lefcheck, J.S., Duffy, J.E., Baker, S.C., Thomson, R.J. *et al.* (2013). Integrating abundance and functional traits reveals new global hotspots of fish diversity. *Nature*, 501, 539 – 542.

Some aspects of the methods need more detail: 1) It is not clear how abundance reductions are applied. Do the authors take random draws from a normal distribution and apply these reductions to some kind of normalised relative abundances?

For abundances reductions, we applied a relatively low level of abundance decrease (i.e. 5% of the total abundance) to obtain rarefaction curves. Testing lower percentages would lead to a very long time computation. In the previous version of the manuscript, there was a mistake in the sentence that referred to this point. This is now corrected (lines 478-479) : the sentence “At each disturbance, we applied a decrease in the abundance of randomly selected species (we recommend to select 5% of the total species richness)” is now corrected as “At each disturbance, we applied a decrease in the abundance of randomly selected species (we recommend to apply a decrease of 5% of the total abundance of the community)”.

2) How does this method deal with highly dimensional functional spaces? How do the authors divide up highly dimensional functional spaces into functional entities?

Thank you also for pointing this critical point. The method is compatible with highly dimensional functional spaces (i.e. more than 2 PCoA axes) since functional entities are defined as multidimensional grid cells or cubes. However, we recommend not using more than 2 axes since considering more axes would scatter species into a

very high number of functional entities (i.e. grid resolution^{number of selected PCoA axes}) resulting in much more cells with only a few or only one species in each of them. In such a case, functional redundancy cannot exist which is not realistic. In the extreme case, every single functional entity would only hold one species, reducing the interest of the framework. This important point is now justified in the methods (l. 443-450) and results (l. 153-155).

Reviewer #2 (Remarks to the Author):

This paper explore the performance of a trait-based framework to assess the vulnerability of biological communities to extinction. For that, they used three working examples of marine communities using past data of species abundances, current presence absence data, and future projections due to climate change. The results show that the trait-based framework, which involves the comparison of observational data with simulated data, is able to document differences in functional vulnerability across the three datasets, although there is an inherent context dependency at both temporal and spatial scales.

I think the aim of this paper is interesting, and goes in line with the current line of thinking that we do not need to know every single mechanistic aspect of the relationship between species characteristics and their vulnerability to predict their responses to environmental changes or perturbations. Moreover, the method has all the ingredients to be adopted by a wide range of ecologist and conservation biologists because it is tractable (the algorithms are well defined), comparable (outcomes can be compared across space and time) and scalable (it can be applied to a wide range of scales).

Thank you for these positive comments stressing the importance and usefulness of our approach. We also thank the reviewer for these helpful comments since our additional analyses revealed a very low sensitivity to arbitrary choices, which then reinforced the robustness of the approach. New results are now included in the revised version of the manuscript (**important : please use the mode 'final version' for line numbers, not use the 'final with tracked changes' view**) (lines 597-621).

As general information, we recomputed vulnerability values both for case studies 1 and 2 because minor mistakes were detected in abundance/occurrence tables from the previous manuscript version, but these modifications did not affect the final patterns.

For the North Sea case study, a *round()* function was applied on abundances values (number of individuals per hour of trawling) prior to vulnerability calculation in the previous manuscript version. The consequence was that the abundance of very rare species for which abundances was inferior to 1 was then converted to zeros, so they were considered as absent from the communities. We thus re-run all analyses (including all sensitivity analyses of the supplementary material) without the *round()* transformation. We did not observe any consistent change.

For the Marine mammals case study, we re-run all analyses because 1 species (on 122) was lacking in the previous manuscript version and used the same traits combinations as Albouy *et al.*, 2020. We did not observe any consistent change.

We also performed additional analyses to quantify the effect of NA in traits data on final vulnerability outputs (including their spatio-temporal patterns), and also tested the impact of traits deletion on final outputs.

Yet, I am confused by one important aspect. The framework is about predicting based on precaution principles. It is about helping decision makers where to put the focus on given time and monetary limitations before the communities are more degraded. However the paper does not present any analysis of the accuracy of the algorithms. It just presents differences in vulnerability across communities, but there is no validation of at least some of these results. In such respect, the authors state in lines 275-277 that the results of the trait framework here presented is congruent with previous studies. In general lines I agree with some of the results presented. Communities in the Northern Hemisphere are more vulnerable compared to other regions (Fig.3), but there are other regions in the world that they are also highly vulnerable (South China sea). Similarly, it seems that the vulnerability based on trait information of the Northern sea has slightly decreased, but to what extent is this also true. In sum, my worry here is that there is not info provided to know to what extent the vulnerability to extinction based on trait info is over or underestimated. The overall temporal or spatial patterns across dataset makes sense but the magnitude is what it remains unclear.

We agree with the reviewer's comment: the magnitude of the vulnerability values would benefit from comparisons (and if possible through quantitative comparisons) with independent data. However, there are neither previous works or independent data sets that can provide such an integrated assessment of vulnerability, which is actually one of the motivating factors underlying the development of our approach. Although spatio-temporal patterns are congruent with previous works, we agree that absolute values would probably not reveal the exact vulnerability (especially for the North Sea fish case study where vulnerabilities are around 90-95%). For that reason, we really think that the most interesting aspect of the proposed framework is the integrative aspect of vulnerability assessment. Moreover, contrary to previous works that also used functional entities, the framework also gives the advantage to both consider functional rarity and the distribution of functional redundancy across the trait space. We considered this point in the discussion to remind the limitations of the presented vulnerability value magnitude (l. 404-407).

My second point is that the method does not distinguish between inherent vulnerability to extinction due to the species characteristics and their abundance from extrinsic vulnerability due to the combination of several perturbations. I can imagine being a decision maker and asking whether it will be effective to focus on a particular community or not. It might be the case that even if I put all the effort in diminishing the impacts and perturbations occurring of a given community its intrinsic vulnerability will be still too high, and therefore, it would make more sense to focus on another communities in which my management could be more effective. I think this point needs to be addressed in order to engage managers and decision-makers. Otherwise, the paper remains a nice but just modeling exercise.

Yes you are right, this point is very important. In the revised version we discuss the concept of risk more deeply: it considers both vulnerability and exposure; fifth assessment report of the IPCC (Field *et al.*, 2014). Here, we only considered the intrinsic vulnerability component through a trait-based and integrative approach. From an ecosystem management perspective, the 'exposure' component should be considered, also in an integrative approach (like in Halpern *et al.*, 2008). Additional sentences are now added in the discussion to remind i) that our metric does not consider the 'exposure' component but only the intrinsic 'vulnerability' component and ii) the importance of including the 'exposure' component for ecosystem management (l. 370-377).

Referred literature:

Field, C. B., Barros, V. R., Mastrandrea, M. D., Mach, K. J., Abdrabo, M. K., Adger, N., . . . Burkett, V. R. (2014). Summary for policymakers. In C. B. Field, V. R. Barros, D. J. Dokken, K. J. Mach, M. D. Mastrandrea, T. E. Bilir, . . . L. L. White, (Eds.) *Climate change 2014: Impacts, adaptation, and vulnerability. Part A: Global and sectoral aspects. Contribution of Working Group II to the Fifth Assessment Report of the Intergovernmental Panel on Climate Change* (pp. 1 – 32). Cambridge, UK: Cambridge University Press.

Halpern, B. S. *et al.* A Global Map of Human Impact on Marine Ecosystems. *Science* 319, 948–952 (2008).

Finally, It is not clear from reading the methods how complete is the trait data. I assume there was no information for all traits for all species. how then the authors proceeded? Did they impute trait data? did they considered NAs? I think this is an important aspect, which leads me more to more general question. How sensitive is the trait framework presented here to have several biases of trait information? I can imagine four sources of biases. 1) Low number of traits, 2) trait information available for some particular communities, 3) trait information available for some particular taxa (families), and trait information obtained for different species across different years. Since the key aim of the paper is based on applying trait info to predictive tools, this must be addressed in the same way that the authors have considered a sensibility analyses based on the number of perturbations.

Thank you for pointing out this critical aspect of our work. As recommended, additional analyses were performed to investigate potential undesired effects of NA and trait selection on final outputs. To do so, we recomputed vulnerability values by removing all species for which a NA was observed for at least one trait and compared these new vulnerability values with those computed from the entire trait dataset (i.e., with all NAs). Additionally, to investigate the impact of NA on final vulnerability values, we also applied our algorithm on fully completed trait tables (i.e., with NAs replaced by estimated values) by using a random Forest technique (*missForest: MissForest R package*; Stekhoven and Bühlmann, 2012) and compared final outputs with initial vulnerability values.

We also performed sensitivity analyses to test for the effect of trait deletion on final outputs. To do so, we re-ran the algorithm for different combinations of traits. For each number of selected traits, we performed 20 iterations. We then compared final outputs to our reference values (coming from the analysis based on all traits) by calculating the correlation coefficients.

All these additional sensitivity analyses reveals that both NAs and trait deletion have a very low effect on final results (i.e. spatio-temporal patterns of the vulnerability; details in the following paragraphs). We then decided to keep the initial results (i.e., where all NAs were kept in traits datasets) but to add more results in the appendix.

For the North Sea fish case study, only 0.7% of the traits table was composed of NA. The proportion of NA was relatively constant during the entire time series and remained particularly low (between 0 and 0.8%; see figure below left). A very low impact of NA has been firstly detected thanks to the analysis performed from the traits dataset without species having at least one NA: the temporal trend of the North Sea fish community is still observable and very close to the initial temporal trend (i.e. decrease of only $1\pm 2\%$ of the vulnerability when using the subsetting traits dataset compared to the entire traits dataset; $r = 0.6$ on absolute values of vulnerabilities and $r = 0.73$ on rankings of vulnerabilities; see the green line in the figure below right). In the same way, we also observed a low impact of NA through the NAs imputation procedure as the temporal trend of the North Sea fish community is still observable: decrease of only $2\pm 2\%$ of the vulnerability by using imputed traits dataset compared to non-imputed traits dataset; $r = 0.55$ on absolute values of vulnerabilities and $r = 0.65$ on rankings of vulnerabilities; see the red line in the figure below right).

We also performed sensitivity analyses to test for the effect of traits deletion on final outputs. To do so, we re-ran the algorithm for each combination of traits from 5 traits to $N-1$ traits ($N = \text{total number of traits}$). For each number of selected traits, we performed 20 iterations. We then compared the final outputs to our reference values (from the analysis based on all traits, i.e. with N traits) by calculating their correlation coefficients. We observed that traits deletion has a low impact on vulnerability estimations as indicated by a very small change in the r correlation coefficient (around 0.7) as the number of selected traits decreases:

For the marine mammals' case study, 5.8% of the traits table was composed of NA. We then investigated potential spatial patterns of NA proportion in traits data. For each site we subsetted the traits table knowing the species pool of the site and computed the proportion of NA in the traits table. At a global scale, the median value of NA proportions was very low ($4.9 \pm 1.7\%$) and no clear spatial pattern was observed:

We also recalculated vulnerability values by deleting species associated with at least one NA in the traits table and compared it with initially computed vulnerabilities (i.e. with all the species considered). By doing so, the number of species considered was reduced from 122 to 70 (which unfortunately led to the deletion of some traits information of all deleted species). As we could expect, we observed substantial deviations from initial vulnerability values (diminution of $13 \pm 35\%$). However, imputing NA led to very low deviations from initial vulnerability values (diminution of $2.7 \pm 10\%$ of vulnerability values after using imputed traits data; correlation with initial vulnerability values: $p = 0.0017$ and $r = 0.75$; see the figure below).

NA effect in case study 2:

Vulnerability from the
non-imputed traits table

Vulnerability from the
imputed traits table

We finally observed that traits deletion has a low impact on vulnerability estimations through the decrease of the r correlation coefficient as the number of selected traits decreases (see figure below). For this particular case study, all these results both highlight that i) NAs must be considered for vulnerability computation, ii) vulnerability values are highly dependent on selected traits, and thus iii) results initially obtained are relevant only for the initial set of selected traits. In order to highlight this important aspect, we added supplementary figures and text (see lines 597-621 + Figure S4 to S10).

For global-scale marine fish communities, the median value of NA proportions was also relatively low ($8.6 \pm 13\%$) and higher NA proportions were -as we could expect- mostly observed in high latitudes:

Despite this latitudinal gradient, the proportion of NAs in the trait database did not significantly affect the magnitude of NA effect on vulnerability values (magnitude = vulnerability computed from all species minus vulnerability computed from species with any NA). These results therefore reinforce that NA values did not affect the final results. Deleting species with at least one NA led to a relatively low increase in the vulnerability (median increase of $0.6 \pm 13\%$). Nonetheless, we think keeping all species for analyses is a better choice than deleting those with NAs in order to keep the trait information we have for these ‘imcomplete’ species.

In the same way, imputing NA in the species traits dataset (thanks to the MissForest technique) also led to very low deviations from initial vulnerability values (increase of $0.04 \pm 3.82\%$ of vulnerability values after using imputed traits data; correlation with initial vulnerability values: $p < 0.001$ and $r = 0.98$; see the figure below).

NA effect in case study 3:

(example on the period [1981-2015])

Vulnerability from the
non-imputed traits table

Vulnerability from the
imputed traits table

All these additional results therefore reinforce that NA values did not affect the final results. We also observed very low impacts of traits deletion on vulnerability computation (see figure below). As for the others case studies, we included new text and supplementary figures for better visibility of NA impacts and traits deletion (see lines 597-621 + Figure S4 to S10).

Referred literature:

Stekhoven D.J., Bühlmann P., 2012. MissForest—non-parametric missing value imputation for mixed-type data. *Bioinformatics* 28(1), 112-118.

As a minor comment, I acknowledge that the paper can not tackle with every single aspect, but I think it worth mentioning in the discussion that the traits considered here do not change across time or species, so the paper ignores the potential effect of trait variation, both genetic (local adaptation) or ecological (phenotypic plasticity) on driving vulnerability differences across communities. If the discussion includes the importance of trait variation within populations, I think it is also worth including the potential of trait variation for the same species across time and space.

Thank you for this comment. This is now acknowledged in the discussion section (l. 377-397).

REVIEWER COMMENTS

Reviewer #1 (Remarks to the Author):

My sentiments about the work remain the same as in the first submission: this is a novel and exciting study that presents and demonstrates a new and generalizable way to quantify a vitally important but wickedly intractable property of ecological communities - functional vulnerability.

I remain unconvinced by some aspects of the framing of the paper. As I thought in the first round of comments (and not really appeased by the response), the random reduction/loss scenarios allow a generalized estimate of functional vulnerability for a community, but it's just a tool to achieve that estimate; much of the framing sells the random scenario as a real asset of this approach, but random scenarios are used widely in ecology to generate null expectations and should not be sold as somehow a novel advance allowing us to deal with uncertainty. On the contrary, in my view the conceptual advance really hinges on the way the authors redistribute species and abundances in trait space to generate best and worse-case communities with which they benchmark functional entity losses in the observed community.

That said, I do not wish to hinder the progress of this work, and as the fundamental research remains outstanding and sure to be impactful, far from it for me to stand in its way.

Reviewer #2 (Remarks to the Author):

I am highly satisfied with the additional analyses that the authors have done to explore the role of NA proportion and trait deletion in modifying observed outcomes. The authors have clearly shown that their species vulnerability assessment is robust to missing information and trait imputation. Overall, I think this manuscript is a very nice contribution showing that functional vulnerability are useful metrics when assessing species vulnerabilities at large scale and for multiple communities.

REVIEWER COMMENTS

Reviewer #1 (Remarks to the Author):

My sentiments about the work remain the same as in the first submission: this is a novel and exciting study that presents and demonstrates a new and generalizable way to quantify a vitally important but wickedly intractable property of ecological communities - functional vulnerability. I remain unconvinced by some aspects of the framing of the paper. As I thought in the first round of comments (and not really appeased by the response), the random reduction/loss scenarios allow a generalized estimate of functional vulnerability for a community, but it's just a tool to achieve that estimate; much of the framing sells the random scenario as a real asset of this approach, but random scenarios are used widely in ecology to generate null expectations and should not be sold as somehow a novel advance allowing us to deal with uncertainty. On the contrary, in my view the conceptual advance really hinges on the way the authors redistribute species and abundances in trait space to generate best and worse-case communities with which they benchmark functional entity losses in the observed community. That said, I do not wish to hinder the progress of this work, and as the fundamental research remains outstanding and sure to be impactful, far from it for me to stand in its way.

We thank the reviewer for highlighting the conceptual advance of our framework and we agree that we didn't give enough importance on the idea to build best and worst-case communities and too much highlight was given to random disturbances.

To consider the reviewer's point, we also expanded the 'Introduction' section (lines 88-100) : "One of the main difficulties in ecosystem health assessment is the need for benchmarks [...] Such conditions or quasi-pristine areas are now virtually absent for most ecosystems on Earth and past data about pre-human conditions are often biased and limited".

We now say:

"Based on species position, species abundances redistribution in the functional trait space and *in silico* simulations of disturbances, our framework is the first one to both consider functional rarity, redundancy and abundances to quantify vulnerability in a multi-threats context" (lines 128-131).

To consider this point, several sentences were added/reworded throughout the manuscript: lines 54-55; 59-60; 127-131; 290-297; 406-408.

In order to give more importance on the framework in itself, we also modified the title of the manuscript, as follows: "A functional vulnerability framework for biodiversity conservation". The word 'precautionary' was then removed from the entire manuscript to give more highlight to the framework in itself. The word 'precautionary' is now replaced by 'integrative' to better illustrate the generalized aspect of the vulnerability estimation.

Reviewer #2 (Remarks to the Author):

I am highly satisfied with the additional analyses that the authors have done to explore the role of NA proportion and trait deletion in modifying observed outcomes. The authors have clearly shown that their species vulnerability assessment is robust to missing information and trait imputation. Overall, I think this manuscript is a very nice contribution showing that functional vulnerability are useful metrics when assessing species vulnerabilities at large scale and for multiple communities.

We thank the reviewer for this positive comment.